# Suppression of β1-Adrenoceptor Autoantibodies is Involved in the Antiarrhythmic Effects of Omega-3 Fatty Acids in Male and Female Hypertensive Rats

**DOI:** 10.3390/ijms21020526

**Published:** 2020-01-14

**Authors:** Barbara Szeiffova Bacova, Jana Radosinska, Gerd Wallukat, Miroslav Barancik, Anne Wallukat, Vladimir Knezl, Matus Sykora, Ludovit Paulis, Narcis Tribulova

**Affiliations:** 1Centre of Experimental Medicine, Institute for Heart Research, SAS, 841 04 Bratislava, Slovakia; barbara.bacova@savba.sk (B.S.B.); miroslav.barancik@savba.sk (M.B.); matus.sykora@savba.sk (M.S.); 2Department of Physiology, Faculty of Medicine, Comenius University, 81108 Bratislava, Slovakia; 3Max-Delbruck Centrum für Molekulare Medizine, 13125 Berlin, Germany; gwalluk@mdc-berlin.de; 4Berlin Cures GmbH, 10719 Berlin, Germany; 5Center of Experimental Medicine SAS, Institute of Experimental Pharmacology and Toxicology, 841 04 Bratislava, Slovakia; vladimir.knezl@savba.sk; 6Institute of Pathophysiology, Faculty of Medicine, Comenius University, 81108 Bratislava, Slovakia; ludovit.paulis@gmail.com

**Keywords:** rats, essential hypertension, autoantibody, lethal arrhythmias, connexin-43, omega-3

## Abstract

The arrhythmogenic potential of β1-adrenoceptor autoantibodies (β1-AA), as well as antiarrhythmic properties of omega-3 in heart diseases, have been reported while underlying mechanisms are poorly understood. We aimed to test our hypothesis that omega-3 (eicosapentaenoic acid-EPA, docosahexaenoic acid-DHA) may inhibit matrix metalloproteinase (MMP-2) activity to prevent cleavage of β1-AR and formation of β1-AA resulting in attenuation of pro-arrhythmic connexin-43 (Cx43) and protein kinase C (PKC) signaling in the diseased heart. We have demonstrated that the appearance and increase of β1-AA in blood serum of male and female 12-month-old spontaneously hypertensive rats (SHR) was associated with an increase of inducible ventricular fibrillation (VF) comparing to normotensive controls. In contrast, supplementation of hypertensive rats with omega-3 for two months suppressed β1-AA levels and reduced incidence of VF. Suppression of β1-AA was accompanied by a decrease of elevated myocardial MMP-2 activity, preservation of cardiac cell membrane integrity and Cx43 topology. Moreover, omega-3 abrogated decline in expression of total Cx43 as well as its phosphorylated forms at serine 368 along with PKC-ε, while decreased pro-fibrotic PKC-δ levels in hypertensive rat heart regardless the sex. The implication of MMP-2 in the action of omega-3 was also demonstrated in cultured cardiomyocytes in which desensitization of β1-AR due to permanent activation of β1-AR with isoproterenol was prevented by MMP-2 inhibitor or EPA. Collectively, these data support the notion that omega-3 via suppression of β1-AA mechanistically controlled by MMP-2 may attenuate abnormal of Cx43 and PKC*-ε* signaling; thus, abolish arrhythmia substrate and protect rats with an advanced stage of hypertension from malignant arrhythmias.

## 1. Introduction

The global incidence of hypertension is growing that represents a serious health problem if not treated properly. The etiology of the disease is multifactorial as shown in humans and in animal models as well [1]. Over time, hypertension leads to chronic inflammation, oxidative stress, autoantibodies (AA) production, structural remodeling as well to alterations in intracellular and intercellular signaling. These events promote development of dilated cardiomyopathy (DCM) and subsequent heart failure (HF) as well as cardiac death due to lethal arrhythmias [2,3].

In this context, it is important that AA directed against the first or second extracellular loop of the adrenergic β1-receptors (i.e., β1-AR) may play a significant role. β1-AA have been found in patients with DCM and myocarditis [4,5,6,7,8,9] as well as in dogs suffering from DCM and in hypertensive rats [10,11]. β1-AA can induce agonistic-like effects on spontaneously beating rat cardiomyocytes, i.e., they are able to activate the adenylate cyclase, increase cAMP production, L-type Ca^2+^ current and action potential duration [10]. In the SHR rats, the β1-AA appear between the 3^rd^ and 5^th^ month after the delivery. The activity could be blocked by the specific β1-AR antagonist bisoprolol but not by the inhibitor of the β2-AR (ICI-118.551) [12]. Permanent increase of sympathomimetic tone ultimately provokes rhythm disturbances mostly due to Ca^2+^ overload [13,14]. Indeed, β1-AA autoimmunity causes idiopathic ventricular arrhythmias in humans [4,6,15] and may contribute to the propensity of hypertensive rat heart to ventricular fibrillation (VF) [2].

Taking into account the pathophysiological role of β1-AA in both HF and life-threatening arrhythmias, the attention is given to the amelioration of β1-AA cardiotoxicity by β1-AR antagonists or immune-adsorption therapy [8,11]. In line with this effort, it is relevant to consider the cardio-protective and antiarrhythmic potential of omega-3 polyunsaturated fatty acids (omega-3) [16,17,18] as a non-invasive and non-pharmacological approach. Omega-3 exerts immuno-modulatory, anti-inflammatory, anti-adrenergic and anti-arrhythmic properties in clinical as well as experimental settings. In animals, protection from lethal arrhythmias was mostly attributed to the attenuation of pro-arrhythmogenic substrate, i.e., abnormalities in myocardial intercellular coupling and extracellular matrix (ECM) [16,19]. The question arises whether omega-3 supplementation could reduce lethal arrhythmia risk via counteracting of β1-AA production. If so, it could provide further evidence for the rationale to monitor and adjust the omega-3 levels in hypertensive heart disease, since this parameter was significantly reduced in male and female spontaneously hypertensive rats (SHR) [20].

With respect to β1-AR, it should be emphasized that matrix metalloproteinases (MMP) that degrade components of ECM also cleavage the extracellular domain of membrane receptors in a variety of pathological processes. MMP-2 is a protease specific to type I and IV collagen, while the latter is an integral part of the cell membrane [21,22]. Interestingly, an increased level of MMP was found in the plasma of hypertensive patients [23] as well as in SHR in the advanced stage of disease [24]. Enhanced degradation of ECM facilitates the occurrence of DCM and HF. Moreover, enhanced cleavage of β1-AR may facilitate AA production and promote arrhythmias. Omega-3 has been shown to suppress MMP activity [24,25]. Taken together, we hypothesize that omega-3 may counteract pro-arrhythmogenic β1-AA production by attenuation of myocardial β1-AR receptors cleavage via inhibition of MMP.

Thus, we focused on the circulating levels of β1-AA alongside with myocardial MMP-2 activity in male and female SHR with an advanced stage of hypertension. Our goal was to elucidate mechanisms of omega-3 involved in protection against lethal arrhythmias. In this context, we examined myocardial subcellular integrity as well as electrical coupling protein, connexin-43 (Cx43) and protein kinase C (PKC) signaling, as established factors affecting arrhythmogenesis that are modulated by β adrenoceptors and omega-3.

## 2. Results

### 2.1. Biometric Parameters of Examined Rats

General characteristics of the 12-month-old experimental animals are summarized in Table 1. Both male and female SHR exhibited significantly higher blood pressure, heart and left ventricular weight compared to healthy animals (WISc) regardless the treatment with omega-3. However, body weight of hypertensive rats was lower.

### 2.2. Incidence of Electrically-Induced Sustained VF

Susceptibility of the heart to lethal VF was tested ex vivo using the Langendorff model. As shown in Figure 1, sustained >2 min lasting VF was induced in about 50% of healthy rats (6 of 10 males and 5 of 10 female rats). In contrast, VF was induced in 100% of males and 80% (8 of 10) in female SHR hearts. Omega-3 intake resulted in suppression of VF incidence in both hypertensive and non-hypertensive rats regardless the gender.

### 2.3. Identification of β1-AA Activity

The β1-AA were identified and characterized with a sensitive bioassay, using cultured spontaneously beating neonatal rat cardiomyocytes. The AA containing immunoglobulins (IgG) were prepared from the sera of the experimental WIS and SHR animals by ammonium sulfate precipitation. It was demonstrated that the IgG prepared from the sera significantly increase the beating rate of the spontaneously beating cardiomyocytes. This positive chronotropic effect of the β1-AA was seen in both male and female SHR but not in the normotensive WISc rats (Figure 2). This activity of the β1-AA was significantly suppressed (> 50%) in SHR treated with omega-3.

### 2.4. Myocardial MMP-2 Activity

Alterations of the extracellular matrix were considered according to the proteolytic activity of MMP-2. Compared to basal MMP-2 activity in the left ventricle of normotensive male and female rats there was a significant increase of this enzyme activity in the myocardium of hypertensive rats of both sexes (Figure 3). Omega-3 intake significantly reduced enhanced MMP-2 activity in male as well as female SHR hearts.

### 2.5. Subcellular Integrity of the Cardiac Cell Membrane (Sarcolemma)

Examination of ultrathin sections from the left ventricles of experimental rats using transmission electron microscopy revealed the degradation of the basement membrane of sarcolemma in male as well as female SHR hearts (Figure 4). Accordingly, lamina lucida and lamina densa that are integral parts of the basal membrane of sarcolemma apposing to extracellular matrix were hardly recognized. In contrast, these layers were well preserved in SHR treated with omega-3. Moreover, non-treated SHR exhibited electron lucent mitochondria (indicating low energized status) compared to electron dense mitochondria (indicating high energized status) of omega-treated SHR.

### 2.6. Topology of Myocardial Cx43

Specific immunolabeling of transmembrane protein Cx43 revealed clear cut differences in its localization in hypertensive male as well as female rat hearts (Figure 5). In contrast to the conventional prevalent distribution of Cx43 at the intercalated disc located gap junctions in non-hypertensive rats, there was enhanced immunopositivity on lateral sides of the cardiomyocytes. Moreover, remarkably disordered distribution of Cx43 was observed in the areas of pronounced extracellular matrix alterations (fibrosis).

### 2.7. Myocardial Cx43 Protein Levels

Western blotting analysis showed that total Cx43 protein and its functional phosphorylated forms were significantly decreased in hypertensive male (Figure 6) and female (Figure 7) rat hearts comparing to non-hypertensive animals. While both parameters were significantly increased in the left ventricle of male as well as female SHR due to omega-3 intake (Figure 6 and Figure 7).

### 2.8. Myocardial PKC-ε Protein Levels

One of the protein kinases which phosphorylates Cx43 is PKC-ε. Western blotting analysis showed that alterations in PKC-ε protein levels paralleled the changes in Cx43 phosphorylated forms. As shown in Figure 8, comparing to non-hypertensive controls PKC-ε was significantly reduced in male and female SHR heart and this decrease was abolished due to treatment with omega-3.

### 2.9. Myocardial PKC-δ Protein Levels

Protein abundance of prohypertrophic and proapoptotic PKC-δ was significantly increased in both male and female SHR hearts when compared to healthy controls (Figure 9), while the upregulation of PKC-δ was significantly attenuated in SHR treated with omega-3.

### 2.10. Desensitization of β1-AR and Effects of MMP Inhibitor and EPA in Cultured Neonatal Cardiomyocytes

Using neonatal cardiomyocytes, we demonstrated desensitization of β1-AR in response to 10 µM isoproterenol (Figure 10A), whilethe desensitization of β1-AR was abolished in presence of MMP inhibitor-GM6001, (10µM) (Figure 10B) or omega-3-EPA (3.3 µM) (Figure 10C).

Isoproterenol-induced desensitization of β1-AR was associated with decrease in their protein levels. The decline of the β1-AR protein level was abolished in the presence of the MMP inhibitor-GM6001 or EPA (Figure 11).

## 3. Discussion

β1-AR are powerful regulators of myocardial contractility and heart function. This is, however, jeopardized by the presence of AA against the β1-AR that promote development of heart failure [8,26] as well as idiopathic ventricular arrhythmias in humans [15]. In line with this, we have shown in this study that both male and female rats suffering from an advanced stage of hypertension exhibit a significant increase of circulating β1-AA and incidence of inducible-VF compared to non-hypertensive animals. Consistent with this, the reduced threshold of VF has also been reported in guinea pigs pre-treated with monoclonal β1-AR antibodies [27], whereas transient ventricular arrhythmias were reported in rats immunized with β1-AA [28].

In addition, hypertensive rats exhibited an increase of myocardial MMP-2 activity, suggesting its possible involvement in malformation of β1-AR resulting in AA production. This view is supported by the fact that omega-3 intake reduced the activity of MMP-2 along with a reduced β1-AA levels as well as with the incidence of VF. Considering β1-AR as an integral part of the cardiomyocyte cell membrane (sarcolemma) it is assumed that their extracellular domain is affected by MMP-2 proteolysis of proteoglycan collagen IV [21,22]. In fact, electron microscopy examination revealed clear cut degradation of collagen IV endowed basal lamina of cardiomyocytes (Figure 4). Such a structural disorder should be associated with functional impairment of sarcolemma [29]. In contrast, treatment with omega-3 resulted in the protection of sarcolemma integrity that might be associated with preservation of β1-AR.

In support of this concept, the experiments on cultured cardiomyocytes showed that the isoproterenol-induced desensitization of β1-AR accompanied by the decrease of their protein abundance was abolished in the presence of MMP-2 inhibitor or EPA. These findings strongly point out to the link between β1-AR desensitization and MMP-2 activity that is affected by omega-3 (EPA). Thus, gain- and loss-of-function experiments in neonatal cardiomyocytes in vitro confirm the idea that suppression of β1-AA by omega-3 might be involved in their antiarrhythmic effects. Furthermore, it was shown that EPA and its Cyp metabolites (P450, EEQ 17, 18) reduced the β1-AR mediated responses to isoproterenol [17,30].

Preservation/recovery of sarcolemma function due to omega-3 intake was further suggested by the attenuation of enhanced pro-arrhythmic localization of Cx43 at the lateral membranes of the cardiomyocyte in hypertensive rat hearts. In this context, it is important to note that β1-AR mediated signaling cascade besides regulation of myocardial contractility at the same time modulate myocardial conduction and direct intercellular electrical and molecular communication ensured by Cx43 channels [31]. Thus, it can be expected that β1-AA might disturb these processes as well.

Indeed, besides the abnormal Cx43 distribution, an increase of β1-AA (i.e., chronic β1-agonistic action) in hypertensive male and female rats was accompanied by a decrease in Cx43 protein level as well as its functional phosphorylated forms at serine368 (Cx43-pS368). Both events have been reported to contribute to malignant arrhythmia propensity by facilitating collapse of both the membrane potential and electrical coupling in various pathophysiological conditions including hypertension [2,16,32,33]. On the other hand, protection against life-threatening arrhythmias was observed by approaches that enhanced Cx43 expression [19] as well as its particular pS368 phosphorylation [24,32,33]. This was also the case in our study, showing that omega-3 intake resulted in an increase of Cx43 protein content and its phosphorylated status.

Concerning the latter, we have shown that protein expression of PKC-ε (one of the protein kinases involved in Cx43-pS368 phosphorylation [34]), reduced in non-treated hypertensive rats, was enhanced due to omega-3 intake. Taken together, this may indicate that myocardial Cx43 dynamically remodels to adapt to sympathetic signaling [31,35], implying potential causative relation between β1-AA and Cx43 expression/phosphorylation. Consequently, it has a fundamental impact on myocardial conduction and susceptibility to arrhythmias [36,37]. Even in patients with DCM the protein level of Cx43 was significantly lower and localized at the lateral sides of cardiomyocytes compared to controls [38]. Interestingly, adverse cardiac Cx43 and MMP-2 responses to the excess of thyroid hormones observed in normotensive rats were blunted in hypertensive animals [24] likely to prevent further deterioration and pro-arrhythmia.

Taken together, it appears that cardiac β1-AR (G_q_-coupled receptors) might be involved in the regulation of Cx43 mediated electrical communication and are probably a critical factor in the maintenance of regular myocardial conduction [35,39], while the desensitization of β1-AR in association with the production of β1-AA may be crucial in the formation of an arrhythmogenic substrate involving aberrant Cx43 signaling as indicates our study. More attention should be paid to this issue when realizing that G_q_-coupled receptors interact with Cx43 at the intercalated disc [39] as well as likely contributing to the occurrence of G_q_-dependent arrhythmias.

Regarding the latter, it should be emphasized that G_q_-coupled receptors also activate PKC signaling in cardiomyocytes [40] and there is functionally important cross-talk between PKC-ε and PKC-δ pathways leading to hypertrophy, apoptosis and fibrosis [41]. We have demonstrated that there was downregulation of PKC-ε and upregulation of PKC-δ in an advanced stage of hypertension, which is characterized by both hypertrophy and fibrosis. PKC-δ stimulates cardiac fibroblast proliferation [42] and chronic PKC-δ overexpression facilitates cardiomyocyte fibrosis, apoptosis and contractile dysfunction [43] whereby very intense G_q_ stimulation induces DCM [40]. In this context, it is important that omega-3 intake normalized PKC-δ as well as PKC-ε expression in hypertensive rats. This action could stimulate beneficial cardioprotective/antiarrhythmic PKC signaling in hypertensive heart disease. As omega-3 certainly exerts pronounced biological actions the precise molecular mechanisms that underlie their actions may not necessarily be as straightforward.

In conclusion, the current study provides an important step forward in promising manipulation of aberrant β1-AR signaling for suppression of cardiac arrhythmias using the cardioprotective potential of omega-3. It is also challenging to translate such an approach to clinical management of adverse β1-AA actions taken into consideration antiarrhythmic effects of omega-3 in human heart diseases [16].

**Limitations of the study:** We did not demonstrate a causal link between the level of β1-AA and incidence of VF rather the implication of β1-AA in VF susceptibility. This idea is worthy of further consideration. Likewise, we did not provide direct evidence about cleavage of β1-ARs by MMP-2 in vivo.

## 4. Materials and Methods

### 4.1. Experimental Animals

The experiments were performed using 10-month-old male and female Wistar rats (WIS, *n* = 64) and age-matched spontaneously hypertensive rats (SHR *n* = 64) that were housed at 23 ± 1 °C, kept at 12-h light/ dark cycles and fed with standard laboratory chow with tap water ad libitum. The maintenance and handling of animals were approved by the State Veterinary and Food Administration of the Slovak Republic, No 289/2003) and performed in compliance with the Ethical Committee of the Institute for Heart Research, Slovak Academy of Sciences according to the European Convention for the Protection of Vertebrate Animals used for Experimental and other Scientific Purposes, Directive 2010/63/EU of the European Parliament.

Half of the animals in each group, WIS and SHR, were treated daily for 2-month with the preparation of omega-3 polyunsaturated fatty acids (omega-3, containing 460 mg eicosapentaenoic (EPA) and 380 mg docosahexaenoic acids (DHA), Pronova BioPharma Norge AS, Norway) at a dose of 200 mg/kg/body weight.

#### Animal Monitoring and Tissue Sampling

Blood pressure was measured at the end of the experiment by tail-cuff plethysmography using the Statham Pressure Transducer P23XL (Hugo Sachs, March-Hugstetten, Germany. Rats were euthanized by intraperitoneal injection of ketamine (100 mg/kg/body weight; Narketan; Vetoquinol UK Ltd., Towcester, UK) and myorelaxant xylazine (10 mg/kg body weight; Xylapan; Vetoquinol UK Ltd., Towcester, UK).

After blood collection, hearts from one part of the animals were quickly excised and weights of the heart (HW), and left ventricle (LVW) were registered. Left ventricle tissue (LV) samples were frozen in liquid nitrogen, stored in a freezer at −80 °C and used for western blot method, gelatin zymography, electron and immunofluorescence microscopy. Another part of the animal hearts was used for the perfusion of isolated hearts in a Langendorff mode.

Blood serum was obtained by blood sample centrifugation at 1200× *g* for 5 min, frozen in liquid nitrogen and stored in a freezer at −80 °C. Serum samples were used for the measurement of serum activity of autoantibodies directed against the β1-adrenergic receptor β1-AA.

### 4.2. Neonatal Rat Cardiomyocytes

The hearts from <3-days-old rats were transferred to phosphate buffered saline (PBS), (animal experiment license number: 411/96, Berlin, Germany). Ventricles were dissected in pieces, washed with PBS containing penicillin/streptomycin (Sigma-Aldrich) followed by PBS only and then suspended in PBS containing 0.2% trypsin. Trypsinization was stopped with heat-inactivated calf serum after 20 min. It was followed by centrifugation for 6 min at 130× *g* and the pellet was transferred to SM20-I medium. For cell count estimation, 100 μL of this suspension was added to 100 μL trypan blue solution. For cell culturing, 2.4 × 10^6^ cells were resuspended in 2.0 mL of complete SM 20-I medium and supplemented with 10% heat-inactivated calf serum, 0.1 mU insulin, and 2 μmol/L fluorodeoxyuridine (Sigma-Aldrich) (to prevent overgrowth of the myocytes by nonmyocytes), transferred to 12.5-cm^2^ Falcon flasks, and cultured as a monolayer for 4 to 8 days at 37 °C. The medium was renewed after the first day and then every second day. Cardiomyocytes began to spontaneously beat after 2 days of culture [44,45].

The first part of the neonatal rat cardiomyocytes was used for analysis the activity of β1-AA. The second part of cells was used for monitoring the functionally processes of β1-AR desensitization. In these experiments, the cells were treated for 120 min with the Isoproterenol (10 µM) (Sigma-Aldrich). After this procedure, the cells were washed with pre-warmed (37 °C) SM20-I and after a new estimation of the basal beating rate the cells were stimulated with Isoproterenol again. These experiments were done with and without the inhibitor of the MMP-GM6001 (10µM) (Millipore) or EPA (Cayman Chemicals). The last part of cells was exposed for 120 min to Isoproterenol (10 µM), Isoproterenol (10 µM) + Matrix metalloproteinase inhibitor (10 µM) and Isoproterenol (10 µM) + Eicosapentaenoic acid (3. 3 µM) and used to measurement protein abundance of β1 receptor. In these experiments, we used 4 days old cardiomyocyte cultures.

### 4.3. Examination of Ventricular Fibrillation Incidence in Langendorff-Perfused Hearts

The hearts of 10 rats from each group were perfused via cannulated aorta in Langendorff mode with oxygenated Krebs–Henseleit solution at a constant pressure of 80 mmHg (1 mmHg = 133.322 Pa) and temperature of 37 °C. After 20 min of heart equilibration, the susceptibility of the heart to electrically-inducible sustained VF (lasting > 2 min) was examined as previously described [46]. Briefly, to induce VF a 1s burst of electrical rectangular pulses at a 35 mA current strength was delivered via stimulating electrodes attached to the epicardium of the right ventricle using Electrostimulator ST-3 (Medicor, Hungary).

### 4.4. IgG Preparation from Rats Blood Serum

IgG were prepared from rats serum by ammonium sulfate precipitation according to Wallukat et al. [45]. Briefly, final concentration of 40% ammonium sulfate and rat serum were incubated for 18 h at 4 °C followed by centrifugation for 15 min at 6000× *g*. The pellet was resuspended in PBS, mixed with saturated ammonium sulfate solution and centrifuged again. The pellet was resuspended in PBS and dialyzed against PBS for 72 h. The dialysis buffer was changed 4 to 5 times. The dialysis membrane (VISKING cellulose, type 27/32, Carl Roth Germany) had a molecular weight cut-off (MWCO) of 14 kDa, ensuring loss of low molecular weight molecules (MW range 170–200 Dalton), including catecholamines. The IgG concentration was adjusted to the serum concentration. No change in subtype composition (IgG 1–4) was observed compared to the original serum. IgG fractions were stored at −20 °C. To register the chronotropic response induced by IgG in rat’s serum the basal beating rate of the cardiomyocytes was recorded in 4–8 days cells cultures [44].

#### Measurement of Autoantibodies Directed Against the β1-Adrenergic Receptor

To measure the activity of β1-AA, a bioassay according to Wallukat and Wollenberger was used [47]. The measurements were done on a heated desk (37 °C) of an inverted microscope. After estimation of the basal spontaneous beating rate of the cardiomyocytes, IgG were added to the cells and incubated for 60 min. The chronotropic response of spontaneously beating cultured neonatal rat cardiomyocytes to the IgG was estimated again. The beating rate of six fields were counted for 15 seconds and average (1 unit of β1-AA activity = 1 beat/min frequency increase; lower limit of detection = 4.0 U; cut off β 1-AA positivity ≥ 8.0 U). Through the use of specific blockers of the β1-AR (bisoprolol hemifumarat, Sigma-Aldrich), or by peptides corresponding to the second extracellular loop of the β1-AR, the cells’ chronotropic response can be attributed to β1-AA [12,45,48].

### 4.5. Gelatin Zymography

MMP-2 activity was assessed in samples from LV tissue as described previously [49]. In short, samples were subjected to sodium dodecyl sulfate-polyacrylamide gels co-polymerized with gelatin (2 mg/mL) followed by electrophoresis. Gels were washed twice with buffer (50 mmol/L Tris-HCl, 2.5% Triton X-100, pH 7.4) and incubated overnight in developing buffer (50 mmol/L Tris-HCl, 10 mmol/L CaCl_2_, 1.25% Triton X-100, pH 7.4) at 37 °C. Staining of gels was performed using 1% Coomassie Brilliant Blue G-250 (dissolved in an aqueous solution containing 10% acetic acid and 40% methanol) and unstaining with the same solution but without dye. MMP-2 activity was seen as white bands on dark background and analyzed using Carestream Molecular Imaging Software (version 5.0, Carestream Health, New Haven, CT, USA).

### 4.6. EM Myocardial Subcellular Integrity

Small, 1–2 mm, LV tissue blocks (five per each heart) were fixed in buffered 2.5% glutaraldehyde, postfixed in 1% OsO4, dehydrated via ethanol series, infiltrated by propylene oxide, embedded in Epon 812 and ultra-thin sections were examined using transmission electron microscopy (Tesla 500, Czech Republic).

The transmural needle biopsies were taken from the free left ventricular wall for transmission electron microscopic examinations. Tissue samples were immediately immersed into ice-cold fixation solution. Its composition was 2.5% glutaraldehyde in 0.1 mol/l sodium cacodylate, pH 7.4. The tissue was then cut into smaller blocks (approximately 1 mm^3^) and fixed for 3 h at 4 °C. This step was followed by washout with cacodylate buffer and the tissue blocks were post-fixed with 1% osmium tetroxide for 1 h at 4 °C. After washout with cacodylate buffer, the tissue was dehydrated in graded series of ethanol, infiltrated with propylene oxide and embedded into Epon 812. Semi-thick sections (1 μm) were cut and stained with toluidine blue for light microscopic examination to choose a representative area. Ultrathin sections were cut using LKB Ultratome, mounted on copper grids and stained with uranyl acetate and lead phosphate prior examination in an electron microscope Tesla 500 [50].

### 4.7. Immunolabeling of Cx43

As previously described [51,52], 10 μm thick cryosections (cryostat Leica CM1950; Leica Biosystems, Wetzlar, Germany) were washed in PBS followed by fixation in ice-cold methanol and permeabilization in 0.3% Triton X-100 and blocked with 1% bovine serum in PBS. Sections were exposed overnight to anti-Cx43 antibody (diluted 1:500, MAB 3068, CHEMICON International, Inc., Temecula, CA, USA) at 4 °C, washed with PBS and exposed to antibodies conjugated with FITC—fluorescein isothiocyanate (diluted 1:500, Jackson Immuno Research Labs, West Grove, PA, USA). After washing with PBS and mounting in the Vectashield (H-1200, Vector Laboratories-Inc., Burlingame, CA, USA) immunostaining was examined in Zeiss Apotome 2 microscope (Carl Zeiss, Jena, Germany).

### 4.8. SDS-PAGE and WB Analysis of Proteins

Protein samples were prepared from neonatal rat cardiomyocytes (4–8 days of cell culture) and from WIS and SHR rats LV tissue for SDS-polyacrylamide gel electrophoresis.

Cultured cardiomyocytes were washed with PBS, scratched off, suspended into 0.1 mL RIPA buffer (PBS, 1% NP40, 0.5% sodium deoxycholate, 0.1% SDS) with inhibitors (µL/mL: 10 mg/mL phenylmethanesulfonyl fluoride 10, aprotinin 30, 100 mmol/L sodium orthovanadate 10) and homogenized by 21-gauge needle. The lysate was incubated on ice for 1 hour and centrifuged for 30 min at 4 °C [53].

LV tissue was powdered in liquid nitrogen and homogenized in SB20 lysis buffer (20% SDS, 10 mmol/L EDTA, 100 mmol/L Tris, pH 6.8).

Tissue as well as cell lysate was diluted in Laemmli sample buffer, boiled for 5 min and an equal amount of protein was loaded for separation by SDS-PAGE on 10% bis-acrylamide gels at a constant voltage of 120 V (Mini-Protean TetraCell, Bio-Rad). Proteins were transfered into a nitrocellulose membrane (0.2 μm pore size, Advantec, Tokyo, Japan), blocked for 4 h with 5% milk in TBS containing 0.1% Tween 20 (TBST). Membrane was incubated overnight with anti-Cx43 antibodies (diluted 1:5000, C6219, Sigma-Aldrich); anti-phospho-serine 368-Cx43 antibodies (diluted 1:1000, sc-101660, Santa Cruz Biotechnology, Dallas, TX, USA); anti-PKC-epsilon antibodies (diluted 1:1000, sc-214, Santa Cruz); anti-PKC-delta (diluted 1:1000, sc-93, Santa Cruz); anti-GAPDH antibodies (diluted 1:1000, sc-25778, Santa Cruz). After washing in TBST the membrane was incubated with a horseradish peroxidase-linked secondary antibody (diluted 1:2000, #7074, Cell Signaling Technology) for 1 h [54]. Proteins were visualised via the electrochemiluminescence method and the density of relevant bands was analyzed using Carestream Molecular Imaging Software (version 5.0, Carestream Health, New Haven, CT, USA).

### 4.9. Monitoring of β1 Receptors Desensitization and Its Possible Reverse

To monitor β1 receptors desensitization, chronotropic response of spontaneously beating 4 to 8 days cultured neonatal rat cardiomyocytes to Isoproterenol was recorded. Six fields with synchronic and rhythmic beating cardiomyocytes on the culture flask were marked. After addition of the Isoproterenol (10 µM), Isoproterenol (10 µM) + Matrix metalloproteinase inhibitor and Isoproterenol (10 µM) + Eicosapentaenoic acid (3. 3 µM) to the culture flasks the beating rate of the cardiomycytes were measured after 5 and 120 min. Then, the cells were washed with prewarmed SM20-I medium and after a renewed estimation of the basal beating rate the cells were incubated again with 10 µM Isoproterenol (130 min) and the beating rate in the six fields were counted again for 15 seconds and average [45,48].

### 4.10. Statistical Analysis

Distribution of variables was examined using the Kolmogorov–Smirnov normality test. One-way analysis of variance (ANOVA) and Bonferroni’s multiple comparison test were used for statistical evaluation. Data expressed as means ± standard error of mean (SEM) were considered as statistically significant when *p* < 0.05. All analyses were carried out with GraphPad Prism version 6.0c (GraphPad Software, San Diego California, USA).

## 5. Conclusions

In conclusion, the current study provides an important step forward in promising manipulation of aberrant β1-AR signaling for suppression of cardiac arrhythmias using the cardioprotective potential of omega-3. It is also challenging to translate such an approach to clinical management of adverse β1-AA actions taken into consideration antiarrhythmic effects of omega-3 in human heart diseases.

## Figures and Tables

**Figure 1 ijms-21-00526-f001:**
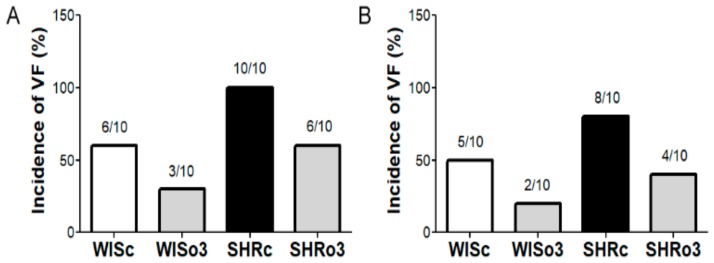
Incidence of ventricular fibrillation (VF) in the male (left panel, (**A**) and female (right panel, (**B**) normotensive and hypertensive rats fed with omega-3. WISc—wistar control rats; WISo3—WISc fed with omega-3; SHRc—spontaneously hypertensive rats; SHRo3—SHRc fed with omega-3. *n* = 10 in each group.

**Figure 2 ijms-21-00526-f002:**
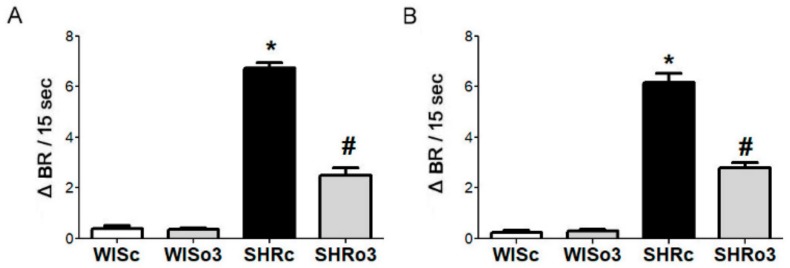
The activity of autoantibodies against the β1-adrenergic receptor recorded as spontaneously beating of cultured neonatal rat cardiomyocytes in response to the immunoglobulins (IgG). The figure represents the difference between the basal beating rate and the increase after stimulation that is suppressed by omega-3. IgG was prepared from an adult male (**A**) and female (**B**) rat blood serum. WISc—Wistar control rats; WISo3—WISc fed with omega-3; SHRc—spontaneously hypertensive rats; SHRo3—SHRc fed with omega-3; sec—seconds; BR— beating rate. *n* = 6 in each group. Data are presented as means ± SEM; * *p* < 0.05 versus WISc; # *p* < 0.05 versus SHRs.

**Figure 3 ijms-21-00526-f003:**
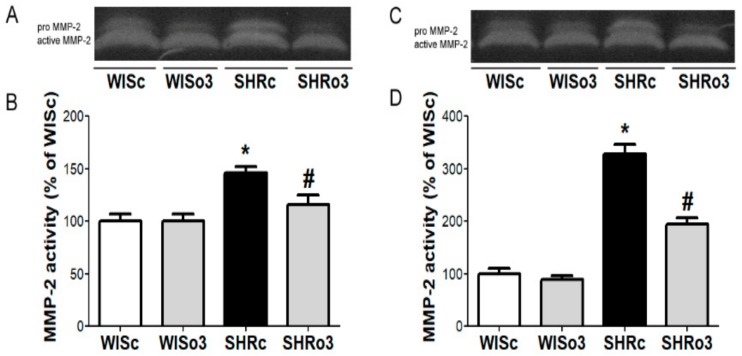
Representative images of zymography (**A**,**C**) and matrix metalloproteinase-2 (MMP-2) activity (**B**,**D**) in the left ventricles of male (left panel, **A**,**B**) and female (right panel, **C**,**D**) normotensive and hypertensive rats fed with omega-3. WISc—wistar control rats; WISo3—WISc fed with omega-3; SHRc—spontaneously hypertensive rats; SHRo3—SHRc fed with omega-3. *n* = 6 in each group. Data are presented as means ± SEM; * *p* < 0.05 versus WISc; # *p* < 0.05 versus SHRs.

**Figure 4 ijms-21-00526-f004:**
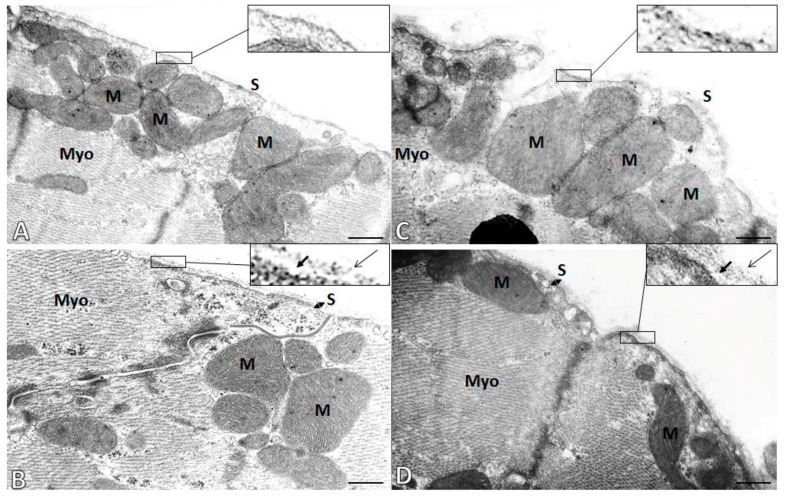
Electron microscopic images demonstrate cardiomyocyte cell membrane (sarcolemma, S) integrity in the male (left panel **A**,**B**) and female (right panel, **C**,**D**) hypertensive rats without (**A**,**C**) or with omega-3 intake (**B**,**D**). Notice the pronounced degradation of the basement membrane of sarcolemma in non-treated male and female rat hearts (**A**,**C**). While preservation of basement membrane containing lamina lucida (short arrows) and lamina densa (long arrows) is seen in omega-3-treated rats (**B**,**D**). M—Mitochondria; Myo—Myofibrils. Scale bar—0.5 µm.

**Figure 5 ijms-21-00526-f005:**
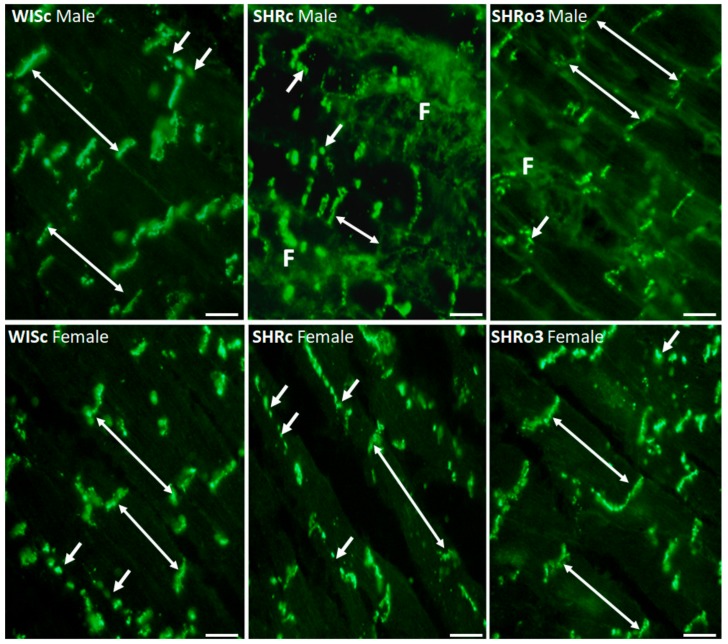
The topology of myocardial connexin-43 (Cx43, green) in the male and female normotensive and hypertensive non-treated and omega-3-treated rats. Note conventional immunofluorescence labeling of Cx43 at the gap junctions (long arrows) and sporadically on lateral surfaces of the cardiomyocytes (short arrows) in normotensive rats, while the pronounced disordered distribution of Cx43 (short arrows) in hypertensive rat hearts was attenuated by treatment with omega-3. WISc—wistar control rats; SHRc—spontaneously hypertensive rats; SHRo3—SHRc fed with omega-3, F—Fibrosis. Objective 40×, Scale bar 10 µm.

**Figure 6 ijms-21-00526-f006:**
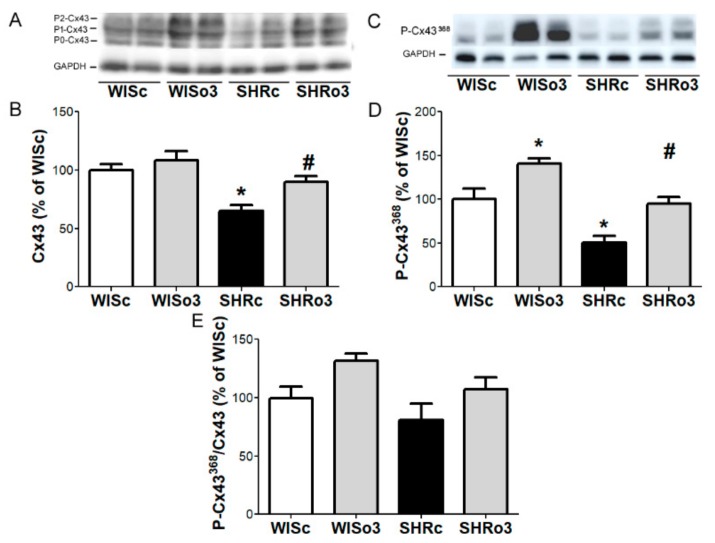
Representative image of western blotting (**A,C**) and protein levels of total myocardial connexin-43 (Cx43, **A,B**), connexin-43 phosphorylated at Serine^368^ (P-Cx43^368^, **C,D**) and ratio of P-Cx43^368^ to Cx43 (**E**) in the left ventricles of male normotensive and hypertensive rats fed with omega-3. Results were normalized to glyceraldehyde-3-phosphate dehydrogenase protein level (GAPDH). WISc—wistar control rats; WISo3— WISc fed with omega-3; SHRc—spontaneously hypertensive rats; SHRo3—SHRc fed with omega-3. *n* = 6 in each group. Data are presented as means ± SEM; * *p* < 0.05 versus WISc; # *p* < 0.05 versus SHRs.

**Figure 7 ijms-21-00526-f007:**
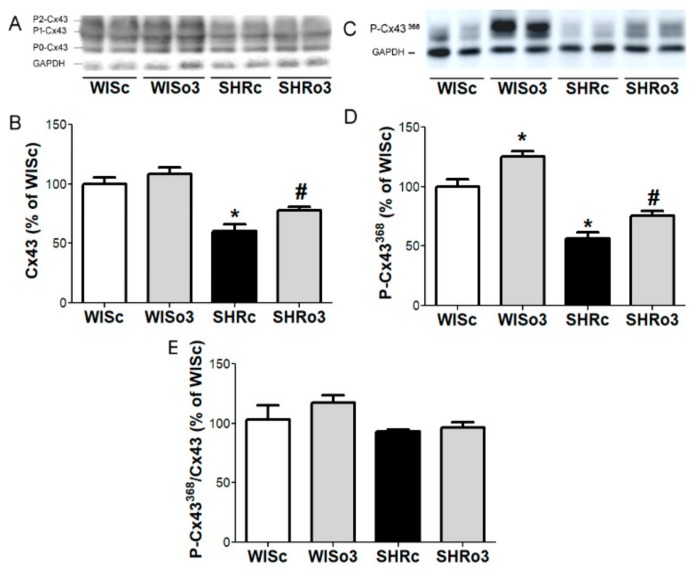
Representative image of western blotting (**A,C**) and protein levels of total myocardial connexin-43 (Cx43, **A,B**), connexin-43 phosphorylated at Serine^368^ (P-Cx43^368^, **C,D**) and ratio of P-Cx43^368^ to Cx43 (**E**) in the left ventricles of female normotensive and hypertensive rats fed with omega-3. Results were normalized to glyceraldehyde-3-phosphate dehydrogenase protein level (GAPDH). WISc—wistar control rats; WISo3—WISc fed with omega-3; SHRc—spontaneously hypertensive rats; SHRo3—SHRc fed with omega-3. *n* = 6 in each group. Data are presented as means ± SEM; * *p* < 0.05 versus WISc; # *p* < 0.05 versus SHRs.

**Figure 8 ijms-21-00526-f008:**
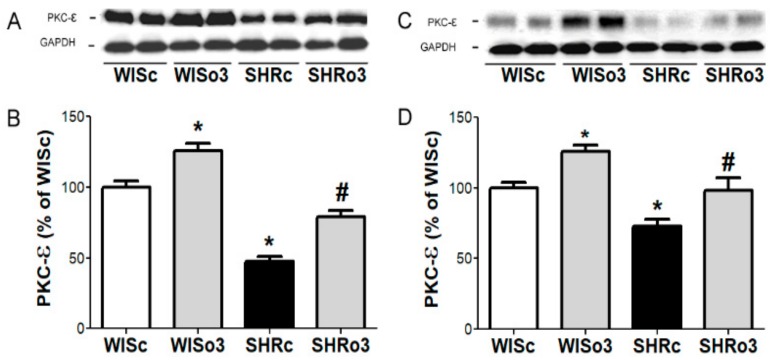
Representative images of western blotting (**A,C**) and protein levels of myocardial protein kinase C-epsilon (PKC-ε, **B,D**) in the left ventricles of male (left panel, **A**,**B**) and female (right panel, **C**,**D**) normotensive and hypertensive rats fed with omega-3. Results were normalized to glyceraldehyde-3-phosphate dehydrogenase protein level (GAPDH). WISc—wistar control rats; WISo3—WISc fed with omega-3; SHRc—spontaneously hypertensive rats; SHRo3—SHRc fed with omega-3. *n* = 6 in each group. Data are presented as means ± SEM; * *p* < 0.05 versus WISc; # *p* < 0.05 versus SHRs.

**Figure 9 ijms-21-00526-f009:**
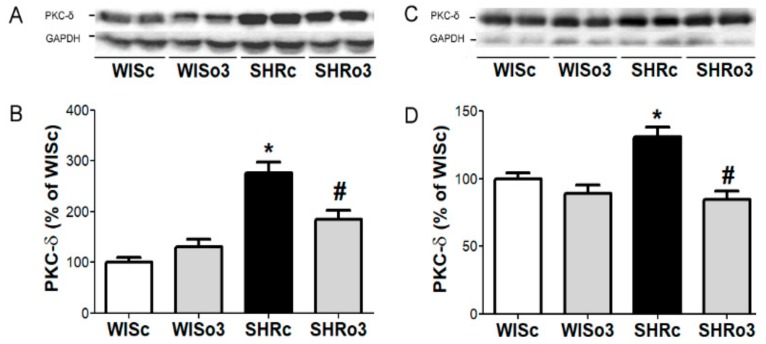
Representative images of western blotting (**A,C**) and protein levels of myocardial protein kinase C delta (PKC-δ, **B,D**) in the left ventricles of male (left panel, **A**,**B**) and female (right panel, **C**,**D**) normotensive and hypertensive rats fed with omega-3. Results were normalized to glyceraldehyde-3-phosphate dehydrogenase protein level (GAPDH). WISc—wistar control rats; WISo3—WISc fed with omega-3; SHRc—spontaneously hypertensive rats; SHRo3—SHRc fed with omega-3. *n* = 6 in each group. Data are presented as means ± SEM; * *p* < 0.05 versus WISc; # *p* < 0.05 versus SHRs.

**Figure 10 ijms-21-00526-f010:**
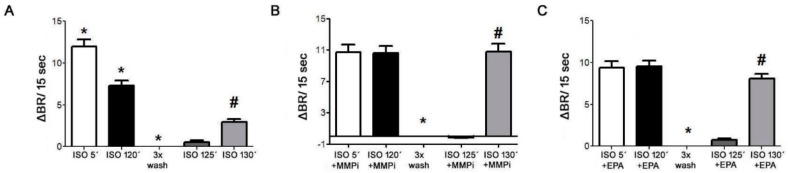
Time response to Isoproterenol (ISO, 10 µM, **A**); Isoproterenol (10 µM) + Matrix metalloproteinase inhibitor (MMPi-GM6001, 10 µM, **B**) and Isoproterenol (10 µM) + Eicosapentaenoic acid (EPA, 3.3 µM, **C**), recorded as spontaneously beating per 15 seconds of cultured neonatal rat cardiomyocytes. After the washing with pre-warmed SM20-I, the basal beating rate was estimated again after stimulation with fresh ISO. BR— beating rate; sec—seconds. Data are presented as means ± SEM; * *p* < 0.05 versus ISO 5´; # *p* < 0.05 versus 3xWasch.

**Figure 11 ijms-21-00526-f011:**
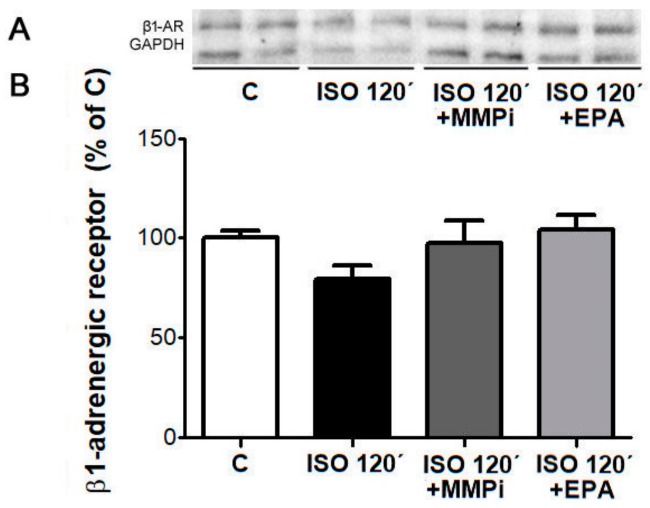
Representative image of western blotting (**A**) and protein levels of β-1 adrenergic receptor (β1-AR, **B**) in cultured neonatal rat cardiomyocytes exposed for 120 min to Isoproterenol (ISO, 10 µM); Isoproterenol (10 µM) + Matrix metalloproteinase inhibitor (MMPi-GM6001, 10 µM) and Isoproterenol (10 µM) + Eicosapentaenoic acid (EPA, 3.3 µM). C—control cells without ISO. Results were normalized to glyceraldehyde-3-phosphate dehydrogenase protein level (GAPDH). Data are presented as means ± SEM.

**Table 1 ijms-21-00526-t001:** General characteristics of experimental rats.

**Variables—Male**	**WISc**	**WISo3**	**SHRc**	**SHRo3**
**BP (mmHg)**	111.74 ± 3.52	95.93 ± 4.47 *	178.77 ± 3.83 *	165.27 ± 3.98
**BW (g)**	467.17 ± 17.95	428.17 ± 6.81	329.50 ± 8.37 *	339.50 ± 6.69
**HW (g)**	1.18 ± 0.04	1.13 ± 0.02	1.41 ± 0.03 *	1.35 ± 0.04
**LVW (g)**	0.81 ± 0.01	0.83 ± 0.02	1.16 ± 0.02 *	1.12 ± 0.03
**Variables—female**	**WISc**	**WISo3**	**SHRc**	**SHRo3**
**BP (mmHg)**	101.18 ± 2.23	98.83 ± 2.91	187.33 ± 7.34 *	172.83 ± 7.02
**BW (g)**	266.83 ± 8.18	295.38 ± 10.05	213.00 ± 5.94 *	216.33 ± 4.97
**HW (g)**	0.83 ± 0.02	0.86 ± 0.02	1.31 ± 0.04 *	1.13 ± 0.04 #
**LVW (g)**	0.59 ± 0.01	0.60 ± 0.01	1.01 ± 0.03 *	0.92 ± 0.06

BP—Blood pressure; BW—body weight; HW—heart weight; LVW—left ventricular weight; WISc—wistar control rats; WISo3—WISc fed with omega-3; SHRc—spontaneously hypertensive rats; SHRo3—SHRc fed with omega-3. *n* = 16 in each group. Data are presented as means ± SEM; * *p* < 0.05 versus WISc; # *p* < 0.05 versus SHRs.

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
