# Peer review of "Suppression of β1-Adrenoceptor Autoantibodies is Involved in the Antiarrhythmic Effects of Omega-3 Fatty Acids in Male and Female Hypertensive Rats"

_ijms, 2020, doi:10.3390/ijms21020526_

Round 1

Reviewer 1 Report

The authors sought to investigate whether omega-3 (EPA, DHA) may inhibit MMP-2 activity to prevent cleavage of β1-AR and formation of β1-AA resulting in attenuation of pro-arrhythmic Cx43 and PKC signaling in the diseased heart. They found that an increase of β1-AA in blood serum of male and female 12-month-old spontaneously hypertensive rats (SHR) was associated with an increase of inducible ventricular fibrillation (VF) comparing to normotensive controls. In contrast, supplementation of hypertensive rats with omega-3 for two months suppressed β1-AA levels and reduced incidence of VF.

This experimental study is well conducted. However, I do have some comments:

Echocardiographic data would be of great interest. Do the authors have information on function al echocardiographic data?

If there is no data available, difference in cardiac biomarkers, e.g. NT-proBNP between groups would be interesting to show.

Please define abbreviations also in the abstract, e.g. ventricular fibrillation (VF)

Overall, I think this is a very interesting study which would be worth publishing.

Author Response

Comments and Suggestions for Authors

The authors sought to investigate whether omega-3 (EPA, DHA) may inhibit MMP-2 activity to prevent cleavage of β1-AR and formation of β1-AA resulting in attenuation of pro-arrhythmic Cx43 and PKC signaling in the diseased heart. They found that an increase of β1-AA in blood serum of male and female 12-month-old spontaneously hypertensive rats (SHR) was associated with an increase of inducible ventricular fibrillation (VF) comparing to normotensive controls. In contrast, supplementation of hypertensive rats with omega-3 for two months suppressed β1-AA levels and reduced incidence of VF.

This experimental study is well conducted. However, I do have some comments:

Echocardiographic data would be of great interest. Do the authors have information on function al echocardiographic data?

If there is no data available, difference in cardiac biomarkers, e.g. NT-proBNP between groups would be interesting to show.

Please define abbreviations also in the abstract, e.g. ventricular fibrillation (VF)

Overall, I think this is a very interesting study which would be worth publishing.

Thank you very much for your time to review our manuscript and we appreciate very much your suggestions.

There is no doubt that echocardiography as well as NT-proBNP assay are relevant approaches to evaluate heart function. In agreement with your suggestion we would like to pay attention to this point in future research, especially in older hypertensive animals. According to our experience when using Langendorff-perfused heart the higher susceptibility of SHR to arrhythmias occurs already in age 3-4 month when heart function is still preserved (index contractility 2049 mmHg/s in SHR versus 2035 mmHg/s in Wistar rats; index relaxation 1224 mmHg/s in SHR versus 1203 mmHg/s in Wistar rats. Benova and Tribulova, Can J Physiol Pharmacol 2013. In this study we focused particularly to explore a possible implication of β1-AA in propensity of hypertensive rat heart to life-threatening arrhythmias.

We defined abbreviations in abstract.

Reviewer 2 Report

The Authors present the well-documented data supporting the beneficial role of omega-3 in protecting rats with hypertension from malignant arrhythmias. The proposed mechanism of this phenomenon involves suppression of β1-AA mechanistically controlled by MMP-2 leading to attenuation of abnormal Cx43 and PKC-ε signaling.

The Authors present the well-documented data supporting the beneficial role of omega-3 in protecting rats with hypertension from malignant arrhythmias. The proposed mechanism of this phenomenon involves suppression of β1-AA mechanistically controlled by MMP-2 leading to attenuation of abnormal Cx43 and PKC-ε signaling. The results are interesting and add to the growing understanding  of the effect of dietary supplementation in omega-3 polyunsaturated fatty acids on cardiac function.

Minor remarks:

Are the changes depicted in Figure 1 statistically significant? Isn’t the title of the manuscript too long? Certain linguistic insufficiencies should be corrected. The manuscript quality could be easily increased by perfecting the style and language. Already the opening sentence is not so fortunate from the linguistic point of view (“The global burden of hypertension is growing and deleterious to health if not properly treated.”). Later in the Introduction “In this context, it is important the fact that AA directed” – it would sound better without “in fact”; “….as well as in dogs suffered from DCM and hypertensive rats [10,11].” should be “dog suffering”. Such small corrections are to be made throughout the text.

Author Response

Comments and Suggestions for Authors

The Authors present the well-documented data supporting the beneficial role of omega-3 in protecting rats with hypertension from malignant arrhythmias. The proposed mechanism of this phenomenon involves suppression of β1-AA mechanistically controlled by MMP-2 leading to attenuation of abnormal Cx43 and PKC-ε signaling. The results are interesting and add to the growing understanding of the effect of dietary supplementation in omega-3 polyunsaturated fatty acids on cardiac function.

Minor remarks:

Are the changes depicted in Figure 1 statistically significant? Isn’t the title of the manuscript too long? Certain linguistic insufficiencies should be corrected. The manuscript quality could be easily increased by perfecting the style and language. Already the opening sentence is not so fortunate from the linguistic point of view (“The global burden of hypertension is growing and deleterious to health if not properly treated.”). Later in the Introduction “In this context, it is important the fact that AA directed” – it would sound better without “in fact”; “….as well as in dogs suffered from DCM and hypertensive rats [10,11].” should be “dog suffering”. Such small corrections are to be made throughout the text.

Thank you very much for your time to review our manuscript and your suggestions. We hope that revision is in accordance with your imagination.

The differences in Fig 1 were not statistically significant and we deleted unappropriated text in legend. We used 10 rats per group while for relevant statistic of incidence of ventricular fibrillation higher number of animals should be used.

We shortened the title as follows: Suppression of β1-adrenoceptor autoantibodies is involved in the antiarrhythmic effects of omega-3 fatty acids in male and female hypertensive rats.

We agree with you that proper linguistic contribute to the quality of the paper. English editing was made by English educated colleague with great experience.

Round 2

Reviewer 1 Report

The manuscript is of good quality, the experimental setup was good, the study is worth to be published.